# Discriminative Evaluation of Sarcopenic Dysphagia Using Handgrip Strength or Calf Circumference in Patients with Dysphagia Using the Area under the Receiver Operating Characteristic Curve

**DOI:** 10.3390/jcm12010118

**Published:** 2022-12-23

**Authors:** Hiroshi Kishimoto, Hidetaka Wakabayashi, Shinta Nishioka, Ryo Momosaki

**Affiliations:** 1Department of Rehabilitation Medicine, Ibaraki Prefectural University of Health Sciences Hospital, Ibaraki 300-0331, Japan; 2Department of Rehabilitation Medicine, Tokyo Women’s Medical University Hospital, Shinjuku-ku, Tokyo 162-0054, Japan; 3Department of Clinical Nutrition and Food Service, Nagasaki Rehabilitation Hospital, Nagasaki 850-0854, Japan; 4Department of Rehabilitation Medicine, Mie University Graduate School of Medicine, Tsu 514-8507, Japan

**Keywords:** sarcopenic dysphagia, handgrip strength, calf circumference

## Abstract

This multicenter cross-sectional study aimed to evaluate the discriminative ability of sarcopenic dysphagia (SD) using handgrip strength (HGS) or calf circumference (CC) in patients with dysphagia. Patients aged 20 years or older who were registered in a database at acute, rehabilitation, long-term care hospitals and home health care facilities were included. Logistic regression analysis was performed using SD as the outcome and HGS, CC, and other confounding factors as covariates, separately by sex. Algorithm-based SD diagnosis and HGS or CC were used as the reference and index tests, respectively. Their accuracy was evaluated using the area under the receiver operating characteristic curve (AUC), and cutoff values were calculated. Of the 460 patients, 285 (126 males) were diagnosed with SD. Logistic regression analysis showed that HGS (odds ratio [OR], 0.909; 95% confidence interval [CI], 0.873–0.947) in males and CC (OR, 0.767; 95% CI, 0.668–0.880) in females were independently associated with SD. The AUC for HGS in males was 0.735 (*p* < 0.001) and CC in females was 0.681 (*p* < 0.001). The cutoff values were 19.7 kg for HGS in males (sensitivity, 0.75; specificity, 0.63) and 29.5 cm for CC in females (sensitivity, 0.86; specificity, 0.48). HGS in males and CC in females provided statistically significant information to discriminate SD from dysphagia.

## 1. Introduction

Sarcopenic dysphagia (SD) is caused by the presence of sarcopenia in systemic and swallowing-related muscles. Sarcopenia is defined as a decreased muscle mass and strength and a consequent deterioration of physical performance [1]. It is a condition that affects the health and quality of life of older adults [2]. Sarcopenia is diagnosed according to the European Working Group on Sarcopenia in Older People [2] and Asian Working Group for Sarcopenia (AWGS) 2019 [3] criteria. In both criteria, muscle weakness, poor physical performance, and loss of muscle mass are incorporated into the condition. SD of the swallowing muscles may enlarge the pharyngeal lumen volume, affecting the pharyngeal swallowing mechanism and function [4]. Furthermore, the mechanism of dysphagia caused by sarcopenia is believed to be secondary to the sarcopenia of the systemic and swallowing-related muscles due to decreased activity, undernutrition, and diseases [5,6,7]. SD is diagnosed using a reliable and validated diagnostic algorithm [8], the criteria of which include the presence of both systemic sarcopenia and dysphagia. Dysphagia without generalized sarcopenia is not considered SD. Moreover, the algorithm excludes cases wherein a neuromuscular disease or other causes of dysphagia other than sarcopenia are present. In this algorithm, handgrip strength (HGS) or normal gait speed is used to assess muscle strength; double X-ray absorptiometry (DXA) or bioimpedance analysis (BIA) is used to assess total body muscle mass; and when DXA or BIA is unavailable, cutoff values for calf circumference (CC) based on previous studies [9] (<34 and 33 cm for males and females, respectively) are to be used. Since the prevalence of SD is high in swallowing rehabilitation-requiring patients [10], institutionalized older adults [11], or patients with dysphagia [12], at 32%, 45%, and 62%, respectively, it is significant to diagnose SD without overlooking it.

The easiest components to measure in the diagnostic algorithm for SD are HGS and CC; however, how low these measurements should be to indicate a high probability of SD remains unclear. Regarding HGS, it has been reported that in a group of female patients with hip fracture, 76.4% of whom had sarcopenia, indicating an association between the degree of HGS loss and dysphagia development [13]. Maeda et al. [14] estimated the cutoff values for HGS of male and female patients for the development of dysphagia after 60 days of hospitalization in a group of patients with restricted oral intake without dysphagia. A study [15] showed that tongue strength was independently associated with HGS in older inpatients in a rehabilitation hospital. Additionally, Wakabayashi et al. [10] showed that patients with SD had significantly less HGS and CC than those without SD. CC is considered a useful indicator for assessing dysphagia in community-dwelling individuals who require long-term care [16]. An independent association between the degree of dysphagia and CC has been found in patients admitted to acute care units [17], and sarcopenia and lower CC have been reported to be strongly associated with dysphagia in male nursing home residents [18]. Furthermore, Kimura et al. [19] noted that CC is a predictor of recovery in patients with dysphagia. However, to date, no study has been conducted on the relationship between SD diagnosis and HGS or CC, and how likely they are to have SD in a group of patients with dysphagia, particularly by sex.

This study aimed to investigate the association between HGS or CC and SD and evaluate the ability of HGS or CC to identify SD in patients with dysphagia, separately by sex.

## 2. Materials and Methods

### 2.1. Study Design and Participants

This study was a cross-sectional study using a multicenter database. The registration data of the Japanese Sarcopenic Dysphagia Database (JSDD), which was created by the Rehabilitation Nutrition Database Committee of the Japanese Association of Rehabilitation Nutrition and the Japanese Sarcopenic Dysphagia Working Group, were analyzed. The database is a web-based registry, and its construction used research electronic data capture (REDCap) [20], developed by Vanderbilt University. The database includes patients with dysphagia aged 20 years or older with a Food Intake LEVEL Scale (FILS) [21] of ≤8 registered at acute care hospitals, convalescent hospitals, long-term care hospitals, nursing homes, and home healthcare facilities (a total of 19 facilities) in Japan. Participating facilities were recruited by the Japanese Association of Rehabilitation Nutrition and the Japanese Working Group on Sarcopenic Dysphagia. The REDCap data input manual was distributed to all data entry personnel at the participating facilities to ensure the accuracy of the data. Data registration took place between November 2019 and March 2021. Detailed characteristics of the patients enrolled in this database are described in a recent paper [12], the following parameters were assessed: age, sex, main disease diagnosed by the 10th Revision of the International Statistical Classification of Diseases and Related Health Problems, whole body sarcopenia diagnosed by the Asian Working Group for Sarcopenia (AWGS) 2019 criteria [3], FILS [21], malnutrition diagnosed by the Global Leadership Initiative on Malnutrition (GLIM) criteria [22], oral status assessed by the Revised Oral Assessment Guide (ROAG) [23] or the Oral Health Assessment Tool (OHAT) [24], ADL assessed by the Functional Independence Measure (FIM) [25] or the Barthel Index (BI) [26], updated Charlson comorbidity index (CCI) [27], C-reactive protein (CRP) and serum albumin levels, dysarthria, hoarseness, aphasia, pressure ulcers, bladder, bowel and kidney function (estimated glomerular filtration rate, eGFR), respiratory status, polypharmacy, number of drugs, and involvement of health care professionals and rehabilitation nutrition teams. Patients enrolled in the JSDD were included in this study, and those without HGS and CC records were excluded.

### 2.2. SD Diagnosis

SD was diagnosed using a reliable and validated diagnostic algorithm [7,8]. This diagnostic algorithm classifies patients into the following three categories: probable, possible, and no SD. The following criteria were required for the diagnosis of SD: the presence of systemic sarcopenia as diagnosed by the AWGS 2019 criteria [3], the presence of dysphagia, and the absence of a causative disease of dysphagia other than sarcopenia. HGS was measured with the AWGS2019 recommended protocol of taking a maximum of at least 2 trials at maximal effort isometric contraction using either both hands or the dominant hand [3]. CC was also measured using the AWGS2019 recommended protocol of taking the maximal value of both calves using a non-elastic tape [3,16]. In this study, the probable and possible categories were collectively regarded as SD.

### 2.3. Statistical Analysis

Statistical analyses were performed using IBM SPSS Statistics version 28 (IBM Corporation, Armonk, NY, USA). For patient characteristics, normally distributed variables were expressed as means ± standard deviations, and non-normally distributed variables were expressed as medians and interquartile ranges. The χ^2^ test, unpaired t-test, and Mann–Whitney U test were used for variable comparisons between groups, as appropriate. To examine the association between SD and HGS or CC, we performed logistic regression analysis using SD or not as an outcome. Age [10,19,28], FILS [10,28], BMI [10,28], BI [10,28], and CCI [27,28] were employed as covariates for confounding adjustment other than HGS and CC, respectively, referring to previous studies. The presence of multicollinearity was confirmed by the correlation coefficient between each covariate. Additionally, SD diagnosis by the diagnostic algorithm was used as the reference standard, and HGS or CC was used as the index test. The accuracy of these tests was evaluated using the area under the receiver operating characteristic (ROC) curve (AUC), and the cutoff value was calculated using the Youden Index. A *p* value of <0.05 was considered statistically significant.

## 3. Results

### 3.1. Patient Characteristics

Of the 467 patients enrolled in the JSDD, 460 (229 males) whose HGS and CC were measured were included in the analysis (Figure 1). Of these, 285 (126 males) were diagnosed with SD. The patient characteristics, separately for males and females, are presented in Table 1. In both sexes, the groups diagnosed with SD were significantly older and had a smaller BMI, and the malnutrition rate according to the Global Leadership Initiative on Malnutrition criteria was also significantly higher. In males, the SD group had a significantly weaker HGS than the non-SD group (12.8 [6.9–19.5] vs. 22.2 [15.5–28.6] kg, respectively), whereas in females, the non-SD group was 8.0 (5.0–13.6) kg and the SD group 9.2 (6.0–12.8) kg, showing no significant difference. CC was significantly lower in the SD group for both sexes: 31.0 (27.7–33.1) and 28 (24.8–30.8) cm in the non-SD and SD groups, respectively for male patients; 28.9 ± 3.8 and 26.5 ± 3.3 cm in the non-SD and SD groups, respectively, for female patients.

### 3.2. Logistic Regression Analysis

Logistic regression analysis showed that HGS in males (Table 2; odds ratio [OR], 0.909; 95% confidence interval [CI], 0.873–0.947; *p* < 0.001) and CC in females (Table 3; OR, 0.767; 95% CI, 0.668–0.880; *p* < 0.001) were independently associated with SD. Confirmation of multicollinearity showed no correlation coefficients above 0.7 as shown in Table 4 and Table 5, but because the correlation coefficient between CC and BMI was as high as 0.618 for men and 0.647 for women, logistic regression analysis was performed excluding BMI from the explanatory variables as a sensitivity analysis. As before exclusion, HGS (OR, 0.909; 95% CI, 0.873–0.947; *p* < 0.001) for males and CC (OR, 0.799; 95% CI, 0.724–0.882; *p* < 0.001) for females were significantly associated with SD.

### 3.3. ROC Curve Analysis

Based on the results of the logistic regression analysis, with SD as the reference standard and HGS for males and CC for females as the index test, the ROC curves for each are shown in Figure 2. The AUC for HGS in males was 0.735 (*p* < 0.001) and that for CC in females was 0.681 (*p* < 0.001); the cutoff values calculated by the Youden Index were 19.7 kg for HGS in males (sensitivity, 0.75; specificity, 0.63) and 29.5 cm for CC in females (sensitivity, 0.86; specificity, 0.48).

## 4. Discussion

In this study, SD was shown to be associated with HGS in males and CC in females. It was possible to identify SD to some extent by HGS in males and CC in females. In addition, logistic regression analysis showed an association with SD in FILS and male age, which is consistent with the results of previous studies [10,28].

For HGS in males, a 19.7 kg cutoff value was sufficient for discriminating SD. Based on the magnitude of the AUC, discriminability was considered moderate. No association with SD was shown for HGS in females. This may be due in part to the fact that female patients in both the SD and non-SD groups had very low HGS, with only 16 of the 232 patients exceeding the AWGS 2019 cutoff value. Nagano et al. [10] showed that in a group of female patients with hip fracture, an HGS of <18 kg increased the risk of developing dysphagia after 1 or 2 weeks of hospitalization. This is consistent with the very low HGS of females in this study, wherein all patients had dysphagia. Maeda et al. [14] found a cutoff value for HGS of <20.2 and 4.1 kg for males and females, respectively, for the development of dysphagia after 60 days of hospitalization in a group of patients with oral intake restriction without dysphagia. For males, the results were considered consistent with the results of this study in a population that had already developed dysphagia. In a study [29] that investigated sex differences in performance in 203 patients with dysphagia of several etiologies using videofluoroscopy, males had more aspiration, laryngeal penetration, residual barium in the hypopharynx and valley, and more upper esophageal sphincter abnormalities than females. The authors stated that the differences may be because of the differences in anatomical and functional swallowing characteristics. A study observing healthy participants [30] showed an anatomical sex difference, with males having significantly larger average pharyngeal structural dimensions in the sagittal plane than females. A study [31] suggested that, functionally, no sex differences are noted in tongue strength; however, sex differences in facial muscle strength exist. The present study showed differences in the functional aspect of sarcopenia, that is, HGS in males, and in the anatomical aspect of sarcopenia, that is, CC in women, depending on whether they were SD or non-SD; however, the cause of this difference remains unclear.

For CC in females, the AUC was 0.681, which is not considered highly discriminatory; however, the 29.5 cm cutoff value allowed some discrimination of SD. No association with SD was shown for CC in males. Yuan et al. [18] reported that low CC was strongly correlated with dysphagia in male nursing home residents. The present study differed from this, possibly because this study included only patients with dysphagia. Kimura et al. [19] reported that after adjusting for sex, a CC of 29.4 cm or greater is a predictor of dysphagia recovery with FILS ≥ 2, which is approximately consistent with the cutoff value for CC for females in the present study. Compared with the values employed in the diagnostic algorithm for SD [3,8,9,32], the cutoff values for CC in females and HGS in males were smaller than those derived from this database. The diagnostic algorithm for SD applied cutoff values for CC and HGS for community residents, which may have resulted in differences in population prevalence from the cutoff values derived from this database, wherein most cases are hospitalized patients.

HGS and CC, which can be assessed in all settings, have been shown to discriminate SD from patients presenting with dysphagia with statistical significance; however, the discriminative power is not high. The prevalence of SD has been reported to be high [10,11,12]. Furthermore, it has been shown that SD is associated with lower swallowing function at discharge in hospitalized patients undergoing dysphagia rehabilitation [10] and that patients with SD show less improvement in swallowing function than those with dysphagia not caused by sarcopenia, even after adjusting for the observation period [28]. In addition to these studies, Wakabayashi et al. [33] noted in their scoping review that a more aggressive nutritional intervention combined with physical rehabilitation to improve muscle strength and swallowing function, that is, multidisciplinary rehabilitation nutrition, may be effective in the treatment of patients with SD. Regarding nutritional management, it has been shown that in patients with SD, nutritional management based on a high provided energy of 30 kcal/IBW/day (kg) or more may significantly improve swallowing ability and provide clinically meaningful improvements in the activities of daily living [34]. Regarding physical rehabilitation, it has been shown that early mobilization by a physical therapist was associated with improved total oral intake in patients with SD after pneumonia [35]. As described by Chen et al. [36] these interventions as early as possible may help prevent the “vicious cycle of sarcopenia and dysphagia”. In patients with dysphagia, prompt screening for SD, even in settings where muscle mass measurements are unavailable, may reduce the risk of overlooking SD by providing easily measured items, including HGS and CC. Furthermore, this may lead to early multidisciplinary rehabilitation nutrition intervention to help improve rehabilitation outcomes for dysphagia.

This study had several limitations. First, we were unable to examine tongue pressure data because of several missing data. Furthermore, because of this, we were not able to make any comparative analysis between probable SD and possible SD groups. In the future, comparative studies between probable SD and possible SD and between probable SD and non-SD should be performed on a patient population with tongue pressure measurement data.

## 5. Conclusions

HGS in males and CC in females provided statistically significant information to discriminate SD from dysphagia. Since HGS and CC can be easily measured in any setting, prompt screening for SD using these measurements may help reduce the risk of overlooking SD and improve rehabilitation outcomes for dysphasia.

## Figures and Tables

**Figure 1 jcm-12-00118-f001:**
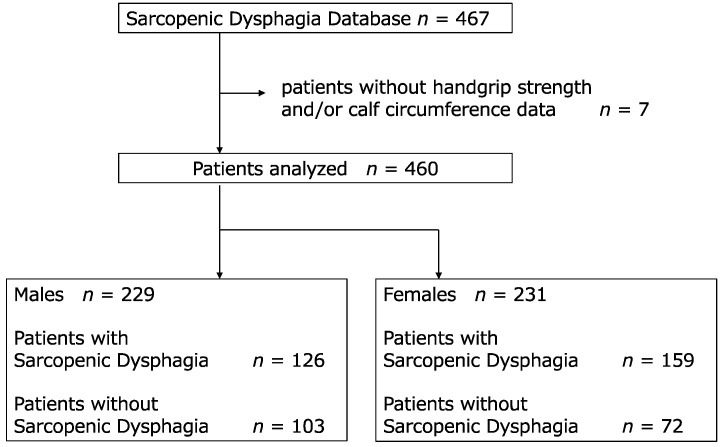
Flowchart of participants.

**Figure 2 jcm-12-00118-f002:**
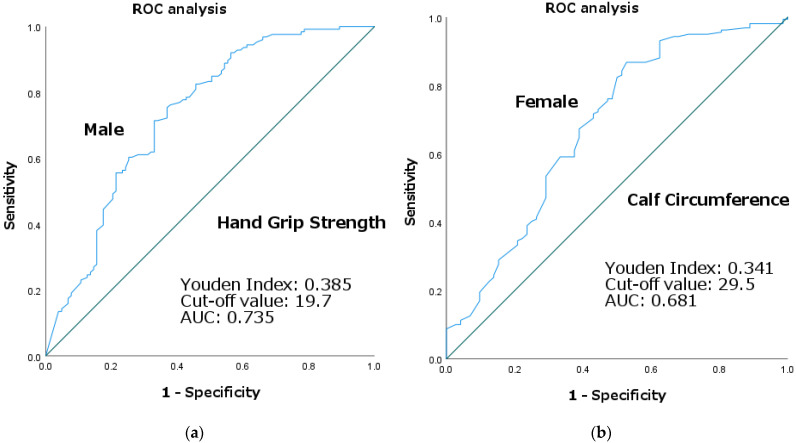
ROC curve analysis of: (**a**) handgrip strength in males, (**b**) calf circumference in females.

**Table 1 jcm-12-00118-t001:** Patient Characteristics. ¶; χ^2^ test, *; Mann–Whitney U test, §; unpaired t-test, SD: sarcopenic dysphagia, BMI: body mass index, HGS: handgrip strength, CC: calf circumference, CCI: Charlson comorbidity index, FILS: Food Intake LEVEL Scale, BI: Barthel index, CRP: C-reactive protein.

	Overall (*n* = 460)	Males (*n* = 229)					Females (*n* = 231)				
		Overall	SD	Non-SD	*p* Value		Overall	SD	Non-SD	*p* Value	
SD, *n*, (%)	285 (62.0)		126					159		0.002	¶
Non-SD, *n*, (%)	175 (38.0)			103					72		
Age (years), median (IQR)	83 (76–88)	80 (71–87)	83 (76–88)	78 (69–83)	<0.001	*	85 (79–90)	86 (82–90)	81 (75–88)	<0.001	*
BMI (kg/m^2^), median (IQR)	19.9 (17.3–22.5)	20.5 (17.7–22.8)	19.9 (16.9–22.4)	20.8 (18.6–23.6)	0.019	*	19.5 (17.2–22.2)	19.2 (17.1–21.3)	20.8 (17.5–24.5)	0.013	*
HGS (kg), median (IQR)	12.0 (6.3–18.8)	17.9 (10.3–24.3)	12.8 (6.9–19.5)	22.2 (15.5–28.6)	<0.001	*	9.1 (5.4–13.0)	9.2 (6.0–12.8)	8.0 (5.0–13.6)	0.796	*
CC (cm), median (IQR)	28.0 (25.1–31.0)	29.4 (26.0–32.0)	28 (24.8–30.8)	31.0 (27.7–33.1)	<0.001	*	27.3 ± 3.6	26.5 ± 3.3	28.9 ± 3.8	<0.001	*
CCI, median (IQR)	2 (1–4)	2 (1–4)	2 (1–4)	2 (0–3.5)	0.221	*	2 (1–4)	2 (1–4)	2 (0–3)	0.487	*
FILS, median (IQR)	7 (3–8)	7 (2–7)	7 (3–7)	6 (1–7)	0.028	*	7 (6–8)	7 (6–8)	7 (1–7)	0.001	*
BI, median (IQR)	25 (5–50)	30 (10–50)	25 (5–45)	30 (10–55)	0.088	*	20 (5–50)	30 (5–50)	17.5 (9–40)	0.198	*
Serum albumin (g/dL), mean ± SD	3.3 (3.0–3.7)	3.4 ± 0.6	3.2 ± 0.6	3.5 ± 0.6	<0.001	§	3.3 (2.9–3.6)	3.2 (2.9–3.6)	3.4 (3.1–3.7)	0.035	*
CRP (mg/dL), median (IQR)	0.7 (0.2–3.2)	0.8 (0.15–3.7)	1.1 (0.3–4.5)	0.7 (0.1–2.5)	0.051	*	0.6 (0.2–2.7)	0.8 (0.3–3.0)	0.3 (0.1–1.2)	0.003	*

**Table 2 jcm-12-00118-t002:** Logistic Regression analysis of sarcopenic dysphagia (males).

	B	Standard Error	Odds Ratio	95% Confidence Interval of Odds Ratio		*p* Value
Age	0.034	0.016	1.035	1.003	1.067	0.032
Food Intake LEVEL Scale	0.144	0.060	1.154	1.026	1.299	0.017
Hand Grip Strength	−0.095	0.021	0.909	0.873	0.947	<0.001
Calf Circumference	−0.059	0.053	0.943	0.850	1.045	0.262
Body Mass Index	0.023	0.053	1.024	0.922	1.136	0.662
Barthel Index	0.009	0.007	1.009	0.996	1.022	0.179
Charlson Comorbidity Index	−0.052	0.086	0.949	0.802	1.123	0.543

**Table 3 jcm-12-00118-t003:** Logistic regression analysis of sarcopenic dysphagia (females).

	B	Standard Error	Odds Ratio	95% Confidence INTERVAL of Odds Ratio		*p* Value
Age	0.028	0.018	1.028	0.993	1.065	0.116
Food Intake LEVEL Scale	0.14	0.07	1.151	1.003	1.320	0.045
Hand Grip Strength	0.015	0.032	1.015	0.953	1.080	0.647
Calf Circumference	−0.266	0.07	0.767	0.668	0.880	<0.001
Body Mass Index	0.054	0.061	1.055	0.937	1.188	0.377
Barthel Index	0.011	0.007	1.011	0.996	1.025	0.139
Charlson Comorbidity Index	0.022	0.092	1.022	0.853	1.224	0.814

**Table 4 jcm-12-00118-t004:** Correlation coefficient table for explanatory variables in logistic regression analysis in males.

Variables		Age	Food Intake LEVEL Scale	Hand Grip Strength	Calf Circumference	Body Mass Index	Barthel Index	Charlson Comorbidity Index
Age	Spearman’s ρ	—						
	*p*-value	—						
Food Intake LEVEL Scale	Spearman’s ρ	0.184	—					
	*p*-value	0.005	—					
Hand Grip Strength	Spearman’s ρ	−0.277	0.091	—				
	*p*-value	<0.001	0.169	—				
Calf Circumference	Spearman’s ρ	−0.286	0.119	0.539	—			
	*p*-value	<0.001	0.073	<0.001	—			
Body Mass Index	Spearman’s ρ	−0.199	−0.033	0.269	0.618	—		
	*p*-value	0.002	0.62	<0.001	<0.001	—		
Barthel Index	Spearman’s ρ	−0.022	0.191	0.507	0.279	0.044	—	
	*p*-value	0.745	0.004	<0.001	<0.001	0.512	—	
Charlson Comorbidity Index	Pearson’s r	0.177	0.113	−0.175	−0.196	−0.129	−0.157	—
	*p*-value	0.007	0.089	0.008	0.003	0.052	0.017	—

**Table 5 jcm-12-00118-t005:** Correlation coefficient table for explanatory variables in logistic regression analysis in females.

Variables		Age	Food Intake LEVEL Scale	Hand Grip Strength	Calf Circumference	Body Mass Index	Barthel Index	Charlson Comorbidity Index
Age	Spearman’s ρ	—						
	*p*-value	—						
Food Intake LEVEL Scale	Spearman’s ρ	0.247	—					
	*p*-value	<0.001	—					
Hand Grip Strength	Spearman’s ρ	−0.104	0.174	—				
	*p*-value	0.116	0.008	—				
Calf Circumference	Spearman’s ρ	−0.177	0.037	0.319	—			
	*p*-value	0.007	0.58	<0.001	—			
Body Mass Index	Spearman’s ρ	0.01	−0.112	0.167	0.647	—		
	*p*-value	0.885	0.089	0.011	<0.001	—		
Barthel Index	Spearman’s ρ	0.102	0.296	0.466	0.266	0.11	—	
	*p*-value	0.123	<0.001	<0.001	<0.001	0.095	—	
Charlson Comorbidity Index	Pearson’s r	0.149	−0.072	−0.15	−0.11	0.003	−0.068	—
	*p*-value	0.024	0.278	0.022	0.096	0.964	0.303	—

## Data Availability

The dataset for this study is not available to the public because of a licensing agreement with the Japanese Society for Rehabilitation Nutrition. The sample dataset is available in a supplement to a previous study by Mizuno et al.

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
