# Peer review of "Discriminative Evaluation of Sarcopenic Dysphagia Using Handgrip Strength or Calf Circumference in Patients with Dysphagia Using the Area under the Receiver Operating Characteristic Curve"

_jcm, 2022, doi:10.3390/jcm12010118_

Round 1

Reviewer 1 Report

This is an interesting study investigating the association of grip strength and calf circumference with sarcopenic dysphagia, and the ability of grip strength or calf circumference to identify sarcopenic dysphagia in patients with dysphagia regarding gender. There are minor points to be amended regarding the English language. Besides, some additional information in the methods section might be useful.

* In the abstract section, and in the conclusion section Lines 225-226, the phrases; "HGS in males and CC in females were statistically significant information for discriminating SD." and "HGS in males was statistically significant information that discriminated SD with moderate discriminative power. CC in females was statistically significant information that discriminated SD, although its discriminative power was slightly low."; it seems that the phrases might be improved to be better understood. The phrase might be "HGS in males and CC in females provided statistically significant information for discriminating SD.".

*In Lines 69, 125, and 185, "shorter CC" might be amended as "lower CC".

*In Line 73, the phrase "and how likely they are to be SD in a group of patients with dysphagia, particularly by sex. " might be amended as "and how likely they are to have SD in a group of patients with dysphagia, particularly by sex."

* In Table 2 and Table 3. Please amend "calf circumfernce" to "calf circumference"

* In the methods and the results section: Please indicate how you chose the variables in the logistic regression model, and explain the significance of the other significant variables.

Author Response

Dear Reviewer 1,

Thank you very much for reviewing our paper and giving it a good evaluation.

* In the abstract section, and in the conclusion section Lines 225-226, the phrases; "HGS in males and CC in females were statistically significant information for discriminating SD." and "HGS in males was statistically significant information that discriminated SD with moderate discriminative power. CC in females was statistically significant information that discriminated SD, although its discriminative power was slightly low."; it seems that the phrases might be improved to be better understood. The phrase might be "HGS in males and CC in females provided statistically significant information for discriminating SD.".

We agree with your suggestion and have changed the sentence in the Summary and Conclusions section to the one you suggested following: HGS in males and CC in females provided statistically significant information to discriminate SD from dysphagia (L28, 259).

* In Lines 69, 125, and 185, "shorter CC" might be amended as "lower CC".

We agree with your suggestion and have corrected “shorter” to “lower” in lines 70 and 146 and “short” to “low” in line 219.

* In Line 73, the phrase "and how likely they are to be SD in a group of patients with dysphagia, particularly by sex. " might be amended as "and how likely they are to have SD in a group of patients with dysphagia, particularly by sex."

We agree with your suggestion and have corrected “be” to “have” in line 74.

* In Table 2 and Table 3. Please amend "calf circumfernce" to "calf circumference"

We agree with your suggestion and have corrected the spelling in Tables 2 and 3.

* In the methods and the results section: Please indicate how you chose the variables in the logistic regression model,

We agree with your suggestion. The explanatory variables were determined by considering the following confounding factors for SD: age, swallowing function status, body size, independence in activities of daily living, and comorbid diseases. Although malnutrition according to the GLIM criteria was a candidate explanatory variable, we did not adopt it because the GLIM criteria encompass assessment of BMI and CC. The following text was inserted in the explanation section of the logistic regression analysis of the method, and the reference numbers of the previous studies referred to were listed for each of the explanatory variables: Age [10,19,22], FILS [10,22], BMI [10,22], BI [10,22], and CCI [22,23] were employed as covariates for confounding adjustment other than HGS and CC, respectively.

and explain the significance of the other significant variables.

We agree with your suggestion and have added the following sentence to Discussion (L190): In addition, logistic regression analysis showed an association with SD in FILS and age, which is consistent with the results of previous studies.

Reviewer 2 Report

Kishomoto and colleagues present an interesting paper that evaluates the use of handgrip strength assessments and calf-circumference assessment as discriminative markers of sarcopenic dysphagia (SD) in dysphagia patients. Overall, this paper was well written and presents valuable clinical findings. However, there are a number of concerns that should be addressed. These are detailed below.

General Comments:

1.       It appears that this manuscript represents a retrospective analysis of data that were collected as part of a larger study (the Japanese Sarcopenic Dysphagia Database), which is certainly an acceptable approach to answering the stated research question. I would just ask that this be clearly defined in the methods.

Specific Comments:

2.       It is currently unclear how handgrip strength and calf circumference were assessed, and standardized, as part of the JSDD protocol. The authors should consider adding this information to the methods.

3.       While the identification of correlation coefficients < 0.7 is generally accepted to rule out multicollinearity between predictor variables, some authors argue that this cut off value may be not be appropriate in all instances (see PMID 27274911 [Vatcheva et al., 2016]). In light of this, the authors may consider providing additional tables and / or text listing the correlation coefficients from the multicollinearity tests for the regression analyses performed in both males and females (tables 2 and 3). This would simply increase the transparency of the analysis.

4.       The authors should consider describing, or referencing, the ROC-AUC ranges used to identify low, moderate, or high discriminatory power.

5.       The lack of discriminatory power for calf circumference in men is an interesting observation, and I’m curious if tissue composition (lean vs. fat mass) would act as a moderator in this case. Perhaps using the thigh skin-fold assessment as an index of leg adiposity could improve the discriminatory power of calf circumference in men. The authors may consider discussing this in the discussion.

6.      In line 212 of the discussion, the sentence beginning with “As described by Chen et al. [31] these interventions as early as possible may help prevent…” should be revised for clarity.

Author Response

Dear Reviewer 2,

Thank you very much for reviewing our paper and giving us a good evaluation.

  1. It appears that this manuscript represents a retrospective analysis of data that were collected as part of a larger study (the Japanese Sarcopenic Dysphagia Database), which is certainly an acceptable approach to answering the stated research question. I would just ask that this be clearly defined in the methods.

We agree with your suggestion and have added a detailed description of the database in the 2.1. Study Design and Participants section with citations from previous studies. In addition, the name of the organization that created the database was corrected on line 82.

  1. It is currently unclear how handgrip strength and calf circumference were assessed, and standardized, as part of the JSDD protocol. The authors should consider adding this information to the methods.

We agree with your suggestion. Both were measured using the protocol recommended by AWGS2019, and we have added that to the 2.2 SD Diagnosis section in lines 112 and 115.

  1. While the identification of correlation coefficients < 0.7 is generally accepted to rule out multicollinearity between predictor variables, some authors argue that this cut off value may be not be appropriate in all instances (see PMID 27274911 [Vatcheva et al., 2016]). In light of this, the authors may consider providing additional tables and / or text listing the correlation coefficients from the multicollinearity tests for the regression analyses performed in both males and females (tables 2 and 3). This would simply increase the transparency of the analysis.

We agree with your suggestion. Tables 4 and 5 were added in lines 173 and 175. As indicated in the tables, the correlation coefficient between CC and BMI was as high as 0.618 for males and 0.647 for females, so logistic regression analysis was performed with BMI excluded from the explanatory variables as a sensitivity analysis. As before exclusion, HGS for males (OR, 0.909; 95% CI, 0.873-0.947; p < 0.001) and CC for females (OR, 0.799; 95% CI, 0.724-0.882; p < 0.001) were significantly associated with SD, and the results of the analysis were confirmed to be robust. This is described in lines 160-166 of the text.
